# Tracking Changes on Soil Structure and Organic Carbon Sequestration after 30 Years of Different Tillage and Management Practices

Ramón Bienes [1], Maria Jose Marques [2,*], Blanca Sastre [1], Andrés García-Díaz [1], Iris Esparza [1], Omar Antón [1], Luis Navarrete [1], José L. Hernánz [3], Víctor Sánchez-Girón [3], María J. Sánchez del Arco [2] and Remedios Alarcón [2]

1   Department of Applied Research and Agrarian Extension, IMIDRA, Finca El Encín. Ctra. A-2, km 38.2, 28800 Alcalá de Henares, Madrid, Spain; ramon.bienes@madrid.org (R.B.); blanca.esther.sastre@madrid.org (B.S.); andres.garcia.diaz@madrid.org (A.G.-D.); iris.esparza@madrid.org (I.E.); omar.anton@madrid.org (O.A.); luis.navarrete@madrid.org (L.N.)
2   Geology and Geochemistry Department, Universidad Autónoma de Madrid, C/Francisco Tomás y Valiente, 7, 28049 Madrid, Spain; maria.jesus.arco@madrid.org (M.J.S.d.A.); remedios.alarcon@madrid.org (R.A.)
3   Department of Forest Engineering, ETSI Montes, Polytechnic University of Madrid, Ciudad Universitaria s/n, 28040 Madrid, Spain; joseluis.hernanz@upm.es (J.L.H.); vsgiron@iru.etsia.upm.es (V.S.-G.)
*   Correspondence: mariajose.marques@uam.es; Tel.: +34-914-974-139

**Abstract:** Long-term field trials are essential for monitoring the effects of sustainable land management strategies for adaptation and mitigation to climate change. The influence of more than thirty years of different management is analyzed on extensive crops under three tillage systems, conventional tillage (CT), minimum tillage (MT), and no-tillage (NT), and with two crop rotations, monoculture winter-wheat (*Triticum aestivum* L.) and wheat-vetch (*Triticum aestivum* L.-*Vicia sativa* L.), widely present in the center of Spain. The soil under NT experienced the largest change in organic carbon (SOC) sequestration, macroaggregate stability, and bulk density. In the MT and NT treatments, SOC content was still increasing after 32 years, being 26.5 and 32.2 Mg ha$^{-1}$, respectively, compared to 20.8 Mg ha$^{-1}$ in CT. The SOC stratification (ratio of SOC at the topsoil/SOC at the layer underneath), an indicator of soil conservation, increased with decreasing tillage intensity (2.32, 1.36, and 1.01 for NT, MT, and CT respectively). Tillage intensity affected the majority of soil parameters, except the water stable aggregates, infiltration, and porosity. The NT treatment increased available water, but only in monocropping. More water was retained at the permanent wilting point in NT treatments, which can be a disadvantage in dry periods of these edaphoclimatic conditions.

**Keywords:** SOC; long-term study; no-tillage; minimum tillage; soil aggregate stability; porosity; available water capacity

## 1. Introduction

Herbaceous crops occupy most of the agricultural surface of the planet, and the choice of soil management practice can significantly modify the quality and physical properties of soil, one of which is the soil organic carbon (SOC) content [1–4], particularly in the context of adaptation to climate change and considering that the accumulation of SOC is a positive indicator of soil conservation. When exploring the maximum potential for SOC sequestration, site-specific conditions must be considered as SOC dynamics vary depending on the edaphic [5] and geographical context. Temperate soils are especially sensitive to the SOC losses associated with some agricultural management systems. Tillage practices can influence SOC storage and dynamics, which is important for both agricultural productivity and carbon sequestration [6,7].

Agricultural soils are usually managed using the 'conventional tillage' (CT) method, based on practices that use the moldboard as the main agricultural implement. However, CT is unsustainable, as it causes reductions in SOC content, destroys soil structure [8,9],

and may reduce its water holding capacity [10]. Soils managed using CT tend to form surface splash crusts and hardened layers at the base of the tillage horizon, hindering water infiltration [11], leading to loss of soil [12] and nutrients [13].

If management practices do not preserve soil quality over time, crop production systems are not sustainable [14]. However, alternative management practices, for example, no-tillage (NT) or direct seeding, minimum tillage (MT), and crop rotations, do exist.

In previous studies, NT compared to CT resulted in increased SOC content [15,16] and improved microaggregate stability [3,17]. The stable microaggregates, with stabilized carbon inside, facilitate its long-term sequestration [15], consequently improving soil properties and promoting greater durability of the newly formed macroaggregates under the NT treatment [18].

NT agriculture is a relatively widely adopted management system that aims to decrease input costs, reduce soil erosion, and sustain long-term crop productivity [19]. It has been demonstrated that soils under NT have higher infiltration, lower bulk density (BD), and larger and/or more continuous pores in the upper profile than those managed using CT methods [20]. Al-Kaisi et al. [21] highlighted the improved conditions of soils managed using NT compared to CT, in terms of increasing SOC storage and stability of micro and macroaggregates. Other improvements in soil parameters such as lower compaction [22], higher soil porosity [23], and water retention increase [24] have also been attributed to the interaction of both tillage systems and crop rotations.

MT is usually defined as a shallow cultivation practice (less than 15–20 cm) performed before the seeding process and over the year to control weeds. In the study area this is the most important management practice, with MT used for 43% of arable land, compared to 15% for CT and 9% for NT [25]. The effect of this management practice on soil properties and crop production in comparison to CT remains unclear [26]. Some studies have demonstrated the effect of MT on the proportion of stable macroaggregates [27,28], bulk density [29], water availability [30], particulate organic matter [31], or SOC content [32]. However, these effects are largely influenced by soil depth and the length of the experiment. Global models of SOC dynamics suggest that changes in SOC due to different management practices may not be detectable until 7–10 years after the beginning of treatment, or even longer [33]. Therefore, when investigating the effect of management practices on soil parameters, the use of long-term experiments is important. The scientific literature on long-term field studies [26,34,35] barely includes semi-arid Mediterranean environments. Therefore, their conclusions are not transferable due to the strong influence of site conditions. In addition, absolute changes can only be determined using the initial SOC content, which is scarcely known in semi-arid agricultural Mediterranean environments [36].

Many studies that are considered as long-term trials usually do not exceed 10 years [6,28,37–45]. There are a few studies conducted over periods between 11 and 20 years [21,27,46–49], and very few studies exceed 20 years [50–52]. Studies exceeding 30 years are extremely rare, for example, Devine et al. [16]. Our study was carried out over 32 years and aimed at analyzing the changes in a Calcic Haploxeralf soil under semi-arid Mediterranean climatic conditions after 32 years (1985–2017) under three different tillage treatments (CT, MT, and NT) and two cropping systems (wheat monoculture and wheat-vetch rotation). The effects of these factors were analyzed to determine the sustainability of crops, and soil quality, based on the following soil physical and chemical properties: SOC, soil BD, porosity, soil water retention, infiltration rate, and saturated hydraulic conductivity.

## 2. Materials and Methods

### 2.1. Study Area

Study plots were located in the municipality of Alcalá de Henares (Madrid, Spain) (40°31′26″ N, 3°18′7″ W) on a plain landscape on a middle terrace of the Henares River. The climate in the region is Continental Mediterranean, with cold winters and hot and dry summers. The average annual temperature is 14.7 °C, the average annual precipitation

is 385 mm (based on recordings taken for 1981–2010) [53], the annual average reference evapotranspiration (ETo) is 1028 mm (2007–2019 period), and the accumulated water deficit during the summer is 643 mm. The soil has a loam dominant texture, a slightly basic pH, and the cation exchange capacity is dominated by calcium. The soil is classified as Calcic Haploxeralf (fine loam family) [54]. At the beginning of the trial, in 1985, a soil pit was opened and the soil horizons were analyzed (Table 1).

**Table 1.** Analytical data from the trial soil pit in 1985. (SOC = soil organic carbon; BD = bulk density).

| Prof. (cm) | Horizon FAO | Texture (USDA) | | | SOC ($g\ kg^{-1}$) | N (%) | C/N | BD ($g\ cm^{-3}$) |
|---|---|---|---|---|---|---|---|---|
| | | Sand % | Silt % | Clay % | | | | |
| 0–18 | Ap | 37.1 | 39.8 | 23.1 | 6.1 | 0.10 | 6.3 | 1.46 |
| 18–34 | ABt | 37.3 | 40.2 | 22.5 | 5.3 | - | - | 1.56 |
| 34–68 | Bt | 31.1 | 33.9 | 35.0 | 4.6 | 0.06 | 6.8 | 1.57 |
| 68–91 | BCt | 43.5 | 32.2 | 24.3 | - | - | - | - |
| 91–131 | Ck | 49.1 | 31.3 | 19.6 | - | - | - | - |

| Prof. (cm) | Horizon FAO | pH (1:2.5 $H_2O$) | Ca $CO_3$ (%) | | K (ppm) | Exchangeable cations ($cmol(+) \cdot kg^{-1}$) | | | |
|---|---|---|---|---|---|---|---|---|---|
| | | | Total | Active | | Ca | Mg | K | Na |
| 0–18 | Ap | 7.8 | 0.0 | - | 27.0 | 14.5 | 3.0 | 0.30 | 0.16 |
| 18–34 | ABt | 7.5 | 0.0 | - | - | 13.8 | 3.6 | 0.10 | 0.20 |
| 34–68 | Bt | 7.3 | 0.8 | - | 19.5 | 12.8 | 4.1 | 0.10 | 0.13 |
| 68–91 | BCt | 7.5 | 1.6 | - | - | 16.6 | 5.0 | 0.15 | 0.22 |
| 91–131 | Ck | 7.8 | 15.6 | 5.8 | - | 19.6 | 5.8 | 0.10 | 0.27 |

### 2.2. Experimental Design

Random blocks were organized on an area of 3.6 hectares, with 18 trial plots of 720 $m^2$ (18 m × 40 m), resulting in the experimental design of three repetitions of 2 rotations × 3 tillage treatments. The study was conducted between 1985 and 2017. Two groups of trials were established (T1 and T2 trials). In each trial, three tillage treatments were tested: CT, MT, and NT with three repetitions (blocks). The T1 trial used winter wheat (*Triticum aestivum* L.) monoculture, while the T2 trial used a wheat-vetch (*Triticum aestivum* L.-*Vicia sativa* L.) rotation. There was no fallow period at any time in any of the plots. Annual fertilization for T1 trial was 80 units (kg) of nitrogen per hectare in the T1 trial and 40 units per hectare in the trial T2. Before drilling, weeds were controlled with a non-selective herbicide (glyphosate) in both trials by adding 1.6 to 2.0 L per hectare depending on the growing of weeds. In the T2 trial, other specific granule post-emergence herbicides based on tribenuron methyl or on bromoxynil were used to control broadleaved weeds.

In the CT treatment, the soil was managed with one pass of moldboard plowing at an average depth of 30 cm, followed by one or two passes with a spring tine cultivator (15–20 cm depth) for sowing preparation. Whenever needed, disk harrowing was applied before moldboard plowing for weed control after harvesting the previous crop. Sowing was performed in early November, after tillage with the chisel. In the MT treatment, the soil was managed with a chisel plow to an average depth of 15–20 cm. Weeds were controlled before tillage, and seedbed preparation and sowing were carried out according to that described for the CT treatment. The NT treatment consisted of sowing with a no-till drill. In all treatments, crops were harvested in early July.

This experiment started in 1985 with the aim to evaluate changes in SOC; originally CT was compared with MT and NT, considering three different rotations systems, wheat monocropping, vetch-wheat, and pea (Pisum sativum)-wheat, the latter was changed to vetch in 2002 [55]. This research follows the same experimental design albeit includes only plots having monocropping and wheat-vetch rotations over time.

*2.3. Soil Sampling and Laboratory Methods*

In 2017, soil samples (composite samples of three subsamples) were randomly taken in each of the plots at two depths: between 0 and 10 cm and between 10 and 20 cm. Soil stratification (SR), calculated as the ratio between the soil parameter values in the topsoil (0–10 cm) and those at 10 to 20 cm depth, was determined for SOC as an index of soil quality [56]. For bulk density (BD) determination, soil cores were taken (with a 100 cm$^3$ cylinder) and subsequently oven-dried until they reached a constant weight.

Samples were air-dried and sieved according to different procedures, to obtain specific-sized fractions to carry out the soil tests described below.

The thirty-six sieved soils (<2 mm) were used for texture analysis using the pipette method [57]; SOC (%) was analyzed using the Walkley and Black method [58]. Following the FAO recommendations [59], we calculated the stock of C (Mg ha$^{-1}$) based on BD and 30 cm thickness, following the Intergovernmental Panel on Climate Change [60], Equation (1).

$$Stock = conc. \times BD \times d \times (1 - \delta 2mm) \times 10^2 \qquad (1)$$

where *Stock* is the stock of C (Mg ha$^{-1}$), *conc.* = C concentration (%), *BD* = bulk density (Mg m$^{-3}$), *d* = soil thickness (m), and *δ2mm* = proportion of gravel (>2 mm), which was negligible in the soils of the study.

Parts per thousand SOC variation over time were calculated using the 4/1000 initiative recommendations [61] (Equation (2)). The 4/1000" Initiative is part of the Global Climate Agenda for Action, promoting scientific cooperation and actions towards reducing greenhouse gas emissions through protecting and increasing SOC stocks yearly at a rate of 4/1000.

$$parts\ per\ thousand = \dfrac{\dfrac{SOC_f - SOC_i}{SOC_f}}{yr} \times 1000 \qquad (2)$$

where $SOC_f$ = final SOC concentration (%), $SOC_i$ = initial SOC concentration (%), and *yr* = years considered for the calculation.

Total nitrogen was determined using the Kjeldahl method [62], available phosphorus content using the Olsen method [63], and exchangeable potassium [64] was measured with atomic emission spectrometry. The water-stable microaggregates (WSA) test was calculated as the percentage of aggregates <2 mm diameter resistant to wet sieving following the Kemper and Rosenau method [65]. Microaggregate samples were submerged and emerged over a 0.25 mm sieve at a rate of 30 oscillations per minute for 3 min. These calculations were corrected for sand content. The WSA test was conducted on three subsamples of 5 g from each of the soil samples.

Sieved soils between 4 and 4.75 mm were analyzed for macroaggregate stability using the CND method (number of drop impacts needed to break aggregates) [66,67]. Thirty macro-aggregates per soil sample were used.

To determine the macro, meso, and microporosity; total porosity; available water capacity (AWC); and BD, 36 undisturbed soil cores were taken (1 sample per treatment × 3 tillage treatments × 3 blocks × 2 rotations × 2 depths). Cores of 5 cm in diameter and 100 cm$^3$ were used. These cores were water saturated via capillarity in a sandbox, and a progressive series of increasing suctions was applied afterward. The sandbox was used for suctions ranging from 0 to 2 pF (0–10 kPa). The Richards' pressure plate system was used for pressures ranging from 2.54 to 4.2 pF (34 to 1500 kPa). Finally, samples were completely oven-dried (24 h at 105 °C) to calculate dry weight and BD. Different pore size groups were established following the classification suggested by Taboada et al. [68,69] and Bienes et al. [70]. These authors separated macropores (pores with diameter > 60 μm; corresponding to suctions between 0.1 and 6.3 kPa), mesopores (pores between 60 and 10 μm; corresponding to suctions between 6.3 and 34 kPa), and micropores (pores ≤ 10 μm; ≥34 kPa). AWC was calculated as the difference between the moisture held in soil microp-

orosity at field capacity (FC, $\geq$2.54 pF) and the moisture at the permanent wilting point (PWP, 4.2 pF).

Infiltration rates were obtained using a simple-ring infiltrometer of 12.7 cm (5 in) in diameter and recording the time necessary to infiltrate 25 mm of water [71] by performing 10 repetitions at the same position. Overall, 18 infiltration tests were carried out (one test per treatment × 3 tillage treatments × 3 blocks × 2 rotations).

### 2.4. Statistical Analysis

The number of repetitions was different for different variables due to the diverse number of tests performed for each measurement. Twelve repetitions were performed for physical-chemical variables and laboratory analysis was in duplicates; six repetitions were performed for variables related to soil porosity due to their little variability, and 3 infiltrations tests were performed per treatment due to operational reasons.

Kruskal-Wallis and Kolmogorov-Smirnov tests, which are based on ranks rather than means, were used as non-parametric alternatives to a one-way variance analysis [72]. The analyses were developed in different stages using Statistica 10 software [73] and PAST 3.2 [74].

## 3. Results

### 3.1. Changes in SOC

Figure 1 shows the evolution of carbon sequestration at depths of 0–10 and 10–20 cm, under different tillage systems since the beginning of the experiment over the elapsed period of 30 years; the figure does not include the rotation effects. Results are shown at 10-year intervals, as this period is considered appropriate for observing soil changes [75]. Data for the first two decades (historic data) were obtained from Hernanz et al. [55]. The SOC in 2016 was higher in the topsoil (0–10 cm), with the highest values obtained under the NT treatment (22.2 Mg ha$^{-1}$). SOC was significantly lower ($p < 0.05$) under the MT (14.4 Mg ha$^{-1}$) and the CT (10.3 Mg ha$^{-1}$) treatments. In the first 10 cm, there was a gradual and significant increase in SOC in both the MT and NT treatments, which was faster in NT as can be observed for the last period of study in Figure 1. However, there were no major changes in the 10–20 cm depth, and there were no significant differences between tillage treatments. SOC managed using CT showed no change over time.

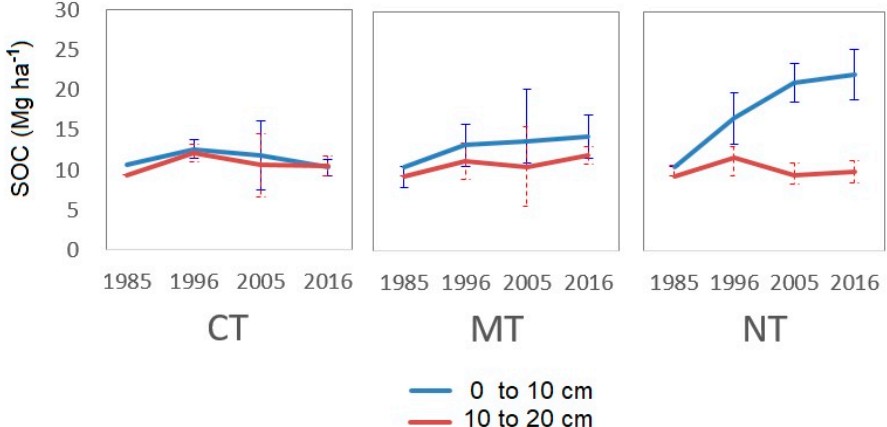

**Figure 1.** Soil organic carbon (Mg ha$^{-1}$, in <2 mm air-dried soil) from 1985 to 2016, under different tillage practices. Historic data from 1984 to 2005 obtained from Hernanz et al. (2009). Data in 2016 are the average of n = 12. CT: conventional tillage, MT: minimum tillage and NT: no-tillage.

Historical data were taken from Hernanz et al. [55], while the 2016–2017 data are new. Under the NT treatment, the SOC stock grew rapidly in the first years. However, in subsequent years, although SOC continued to increase, the rate of growth slowed. Under the MT treatment, the response was slower, and over the last 20 years, the increase in C

stock was almost stable. Considering the 0 to 20 soil thickness, there was a significant increase in SOC stock as the sum of the two layers (0 to 10 and 10 to 20 cm) ranged from 20.8 Mg ha$^{-1}$ in CT, compared to 26.5 in MT and 32.2 in NT, however, increases in SOC were only attributed to the topsoil (0–10 cm depth) as variations in the 10–20 cm depth were negligible. Consequently, over the 32 years, soil management (tillage and crop rotation factors) led to a stratification (SR) in the topsoil. The SR was more evident under the NT treatment, and the SOC content was particularly affected (Table 2). The SR revealed the effect of tillage intensity on SOC accumulation (Table 2). This soil parameter was higher in the 0–10 cm soil depth compared to the 10–20 cm depth, and the accumulation in the topsoil gradually increased as the tillage intensity decreased (CT < MT < NT). Furthermore, regardless of the type of tillage, under the wheat-vetch rotation, there was an accumulation of SOC in the upper 10 cm of soil.

**Table 2.** Stratification relationships (SR: contents at 0–10 cm/contents at 10–20 cm) of SOC, total nitrogen (N), and available phosphorus (P) and potassium (K) under different tillage types and crop rotations. Data obtained from the last two years of the experiment.

| | Factors | | | | |
| --- | --- | --- | --- | --- | --- |
| | Tillage (n = 12) | | | Crop rotation (n = 18) | |
| | **CT** | **MT** | **NT** | **Wheat-Wheat** | **Wheat-Vetch** |
| SOC 0–10 cm (g kg$^{-1}$) | 7.5 ± 0.7 [c] | 10.6 ± 1.9 [b] | 15.4 ± 2.2 [a] | 10.8 ± 3.5 [a] | 11.5 ± 3.9 [a] |
| SOC 10–20 cm (g kg$^{-1}$) | 7.5 ± 0.9 [ab] | 7.8 ± 0.7 [a] | 6.6 ± 0.9 [b] | 7.2 ± 0.9 [a] | 7.4 ± 1.0 [a] |
| SR SOC | 1.01 | 1.36 | 2.32 | 1.50 | 1.55 |
| N 0–10 cm (g kg$^{-1}$) | 0.95 ± 0.11 [b] | 1.13 ± 0.16 [b] | 1.49 ± 0.21 [a] | 1.11 ± 0.26 [a] | 1.27 ± 0.34 [a] |
| N 10–20 cm (g kg$^{-1}$) | 0.82 ± 0.08 [a] | 0.82 ± 0.15 [a] | 0.76 ± 0.14 [a] | 0.77 ± 0.13 [a] | 0.83 ± 0.12 [a] |
| SR N | 1.16 | 1.38 | 1.96 | 1.44 | 1.53 |
| P 0–10 cm (mg kg$^{-1}$) | 15.5 ± 4.2 [b] | 21.8 ± 6.8 [a] | 24.2 ± 5.6 [a] | 18.9 ± 5.7 [a] | 22.1 ± 7.3 [a] |
| P 10–20 cm (mg kg$^{-1}$) | 16.8 ± 4.6 [a] | 15.08 ± 3.8 [a] | 13.4 ± 4.9 [a] | 14.1 ± 3.7 [a] | 16.1 ± 5.2 [a] |
| SR P | 0.92 | 1.45 | 1.80 | 1.34 | 1.37 |
| K 0–10 cm (mg kg$^{-1}$) | 308 ± 99 [b] | 357 ± 109 [ab] | 428 ± 94 [a] | 413 ± 118 [a] | 316 ± 78 [b] |
| K 10–20 cm (mg kg$^{-1}$) | 255 ± 58 [a] | 248 ± 68 [a] | 241 ± 69 [a] | 277 ± 48 [a] | 219 ± 65 [b] |
| SR K | 1.21 | 1.44 | 1.78 | 1.49 | 1.44 |

CT: conventional tillage, MT: minimum tillage and NT: no-tillage. Different letters indicate statistically significant differences between treatments according to Tukey's test ($p < 0.05$).

Similarly, an increase of N was observed in NT treatments at 0–10 cm depth, in this case, the SR indicated that N content at the topsoil nearly doubled the value found at 10 to 20 cm depth (SR = 1.96). The effect of rotation also showed an increasing trend although variability impeded statistical significance.

The content of P and K followed the same pattern: higher concentration at the surface (0–10 cm) and higher concentration as tillage intensity decreased. This pattern yielded progressively higher SR for P and K due to tillage treatments, but it was not detected for rotation treatments.

### 3.2. Changes in Physical and Chemical Properties

In order to understand the combined effects of rotation and tillage, changes in soil parameters are collated in Table 3, in which soil depths between 0 and 20 cm are considered together. Soils managed using monocropping methods are more sensitive to tillage treatments. The CT treatment significantly decreased aggregate stability, SOC, TN, and P, and especially noticeable was the decreased AWC. On the contrary, CT increased macroporosity and mesoporosity and, therefore, decreased BD compared to the other two tillage systems. On average, there were no significant differences in total porosity between treatments.

**Table 3.** Results of Kolmogorov-Smirnov test, showing the influence of tillage and rotation on plots with wheat-wheat monocropping and a wheat-vetch rotation in 2015 and 2016, after 32 years of treatments in 0 to 20 cm of soil thickness. Significant differences are marked at $p < 0.05$. Average values and standard deviation are stated. Different letters in each row indicate differences between different tillage types for the same cultivation system (monocrop or rotation); any differences found between rotation systems under the same tillage system are marked by an asterisk (*).

| Soil Parameters | n | 2 Factors (Rotation and Tillage System) | | | | | |
|---|---|---|---|---|---|---|---|
| | | Wheat-Wheat (Monocrop) | | | Wheat-Vetch (Rotation) | | |
| | | CT | MT | NT | CT | MT | NT |
| CND | 12 | 10.0 ± 2.64 [b] | 12.7 ± 6.14 [ab] | 17.4 ± 8.7 [a] | 13.9 ± 7.2 [a] | 16.1 ± 12.6 [a] | 18.3 ± 10.6 [a] |
| WSA (%) | 12 | 19.1 ± 8.8 [a] | 26.1 ± 7.6 [a] | 24.2 ± 8.3 [a] | 24.3 ± 9.5 [a] | 28.4 ± 11.7 [a] | 25.9 ± 8.8 [a] |
| Inf (mm h$^{-1}$) | 3 | 84 ± 19 [a] | 145 ± 128 [a] | 69 ± 24 [b] | 126 ± 38 [a] | 105 ± 20 [a] | 44 ± 27 [b] |
| SOC (g kg$^{-1}$) | 12 | 7.2 ± 0.6 [b] | 9.1 ± 1.8 [a] | 10.7 ± 4.5 [ab] | 7.8 ± 0.9 [a] | 9.3 ± 2.3 [a] | 11.3 ± 5.1 [b] |
| TN (%) | 12 | 0.08 ± 0.01 [b] | 0.09 ± 0.02 [a] | 0.11 ± 0.04 [a] | 0.09 ± 0.4 [b] | 0.10 ± 0.02 [a] | 0.12 ± 0.05 [a] |
| P (mg kg$^{-1}$) | 12 | 14.1 ± 2.6 [b*] | 18.8 ± 5.6 [a] | 16.7 ± 6.3 [ab] | 18.2 ± 4.9 [a*] | 18.1 ± 7.4 [a] | 20.9 ± 8.2 [a] |
| K (mg kg$^{-1}$) | 12 | 305 ± 89 [a] | 343 ± 113 [a] | 388 ± 125 [a] | 258 ± 75 [a] | 262 ± 82 [a] | 281 ± 103 [a] |
| BD (Mg m$^{-3}$) | 6 | 1.36 ± 0.10 [b] | 1.45 ± 0.11 [ab] | 1.51 ± 0.08 [a] | 1.42 ± 0.10 [a] | 1.46 ± 0.16 [a] | 1.45 ± 0.11 [a] |
| Macrop (% *v/v*) | 6 | 10.9 ± 3.7 [a*] | 8.4 ± 4.4 [ab] | 5.2 ± 2.1 [b] | 4.6 ± 2.1 [a*] | 5.2 ± 5.3 [a] | 4.1 ± 3.6 [a] |
| Mesop (% *v/v*) | 6 | 15.5 ± 0.7 [a*] | 12.8 ± 2.1 [ab*] | 12.5 ± 1.4 [b] | 20.5 ± 3.4 [a*] | 19.1 ± 4.6 [a*] | 18.8 ± 4.0 [a] |
| TP (% *v/v*) | 6 | 49.1 ± 3.6 [a] | 47.4 ± 5.2 [a] | 44.0 ± 2.3 [a] | 48.9 ± 4.8 [a] | 47.1 ± 6.0 [a] | 46.5 ± 6.1 [a] |
| FC (% *v/v*) | 6 | 22.7 ± 1.2 [b] | 26.2 ± 2.5 [a] | 26.3 ± 1.2 [a] | 23.7 ± 1.6 [a] | 22.8 ± 3.4 [a] | 23.6 ± 3.9 [a] |
| PWP (% *v/v*) | 6 | 8.1 ± 0.5 [b] | 10.6 ± 1.9 [a] | 9.7 ± 0.9 [a] | 8.9 ± 1.0 [a] | 9.0 ± 1.4 [a] | 10.6 ± 1.3 [a] |
| AWC (% *v/v*) | 6 | 14.6 ± 1.2 [b] | 15.6 ± 0.9 [ab] | 16.6 ± 0.6 [a*] | 14.9 ± 1.1 [a] | 13.8 ± 2.4 [a] | 13.0 ± 2.8 [a*] |
| AWC/TP (% *w/w*) | 6 | 29.7 ± 2.6 [b] | 33.4 ± 4.9 [ab] | 37.8 ± 3.0 [a*] | 30.8 ± 4.9 [a] | 29.8 ± 7.1 [a] | 28.0 ± 5.9 [a*] |

CT = Conventional tillage, MT = minimum tillage and NT = no-tillage, crop rotation and depth. CND = counting number drop, WSA = water stable aggregates, Inf = infiltration rate, SOC = soil organic carbon, TN = total nitrogen, P = phosphorus (Olsen), K = available potassium, Macrop = macroporosity, Mesop = mesoporosity, Microp = microporosity, TP = total porosity, FC = field capacity, PWP = permanent wilting point, AWC = available water capacity, BD = bulk density, n = number of sampling.

The wheat-vetch rotation seemed to mitigate the effects of tillage treatments documented above. There were then no differences in aggregate stability or BD between tillage methods when rotation wheat-vetch was performed.

When comparing the tillage treatment under different rotation systems (different capital letters in Table 3), we found that rotation produced higher mesoporosity, which ranged from 12.5 to 15.5 under monocropping and 18.8 to 20.5 under rotation. No differences were found for nutrients due to rotation.

The intensity of tillage progressively increases from NT to MT and CT; this rank is reflected in the results of the variables studied, and the MT methods usually showed intermediate values between NT and CT.

From the methodological point of view, the changes in microaggregate stability were not detected using the WSA method.

### 3.3. Changes in Water Infiltration and Water Availability

The average infiltration measured using a single ring infiltrometer was 96 mm h$^{-1}$. No significant differences were noticed between the CT and MT treatments, or between crop rotations (wheat-wheat, wheat-vetch). However, under the NT treatment, there was less water infiltration in soils.

With regard to AWC, the CT treatment reduced significantly this parameter compared to NT in monocropping (14.6% compared to 16.6% respectively). Rotation with vetch reduced even more this AWC although only for NT treatment (16.6% in W-W compared to 13% in W-V). As mentioned before, total porosity was similar between treatments, but when considering the ratio AWC/TP, there was a significant decrease in soils managed using CT and monocropping; this ratio was less than 30% for CT treatment, compared to 33.3 and 37.8% in MT and NT respectively. Therefore, this management practice reduced the ability of soil to hold water as the AWC/TP ratio provides information about the percentage of TP that is occupied by available water for crops. There were then no differences in

AWC between CT and MT methods when rotation wheat-vetch was performed. The water retention curves of soils under different treatments (tillage and rotation) can be observed in Figure 2. On the left, soils under monocropping experienced significant differences between porosity when soils were not wet; these differences were only noticed between NT and CT, as is also indicated in Table 3. Particularly, the little variations in porosity for the NT treatments were noted. On the right, soils under rotation showed similar values of porosity, and therefore, similar AWC, with no significant differences between tillage treatments.

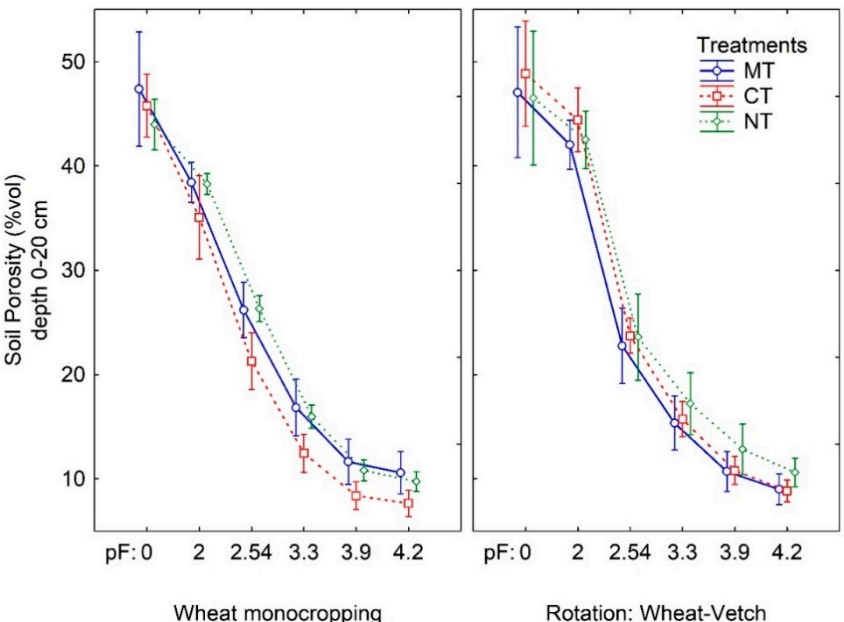

**Figure 2.** Water retention curves of soils considering 0 to 20 cm depth and different tillage treatments, conventional tillage (CT), minimum tillage (MT) and no-tillage (NT) and rotations.

## 4. Discussion

There were no significant differences in crop yield between treatments (average $2.5 \pm 1.3$ Mg ha$^{-1}$, Hernanz et al. [76]). However, several changes were observed in soil parameters after 32 years of different management practices.

### 4.1. Changes in SOC

In 1985, the soils had 10.6 Mg ha$^{-1}$ of SOC in the upper 10 cm layer and 9.4 Mg ha$^{-1}$ in the 10 to 20 cm layer (Figure 1). Considering the upper 20 cm layer of soil, the SOC stock in 2017 had reached an average of 20.8 Mg C ha$^{-1}$ under CT management, 26.5 Mg C ha$^{-1}$ under MT management, and 32.2 Mg C ha$^{-1}$ under NT management. This implies an annual average increase of 10 per thousand under MT management and 14 per thousand under NT management over the 32 years considered. The progression of carbon accumulation suggests that these soils have not yet reached their maximum accumulation potential or their stabilization curve. However, the accumulation rate decreased after 11–12 years [55].

In the upper 20 cm, SOC under the CT treatment was approximately 7 g kg$^{-1}$, which is below the limit of 10 g kg$^{-1}$ considered typical of poor soils that are vulnerable to desertification [77]. Under the NT system that preserves soils, the current carbon stock is 32.2 Mg C ha$^{-1}$, which is higher than the value reported by Rodríguez Martín et al. [78] for central Spain, but much lower than the average value for Spain (56 Mg C ha$^{-1}$, ranging from 20 Mg C ha$^{-1}$ in the center and south of the country, and 150 Mg ha$^{-1}$ in the cooler and wetter northern zones). The low initial rates of SOC in the topsoil (0–10 cm depth), may explain the high carbon sequestration figures found in this updated study, which shows that the initial value has doubled over the last 32 years of NT management. As expected, the greatest increases in SOC were observed in the topsoil, under the NT and MT treatments, although in the latter case the changes were not as pronounced. This may be a consequence

of the accumulation of plant litter in the topsoil. Continuous carbon sequestration for more than 20 years may be common in certain conditions such as those described in this study and others in the literature. For example, Minasny et al. [79] reported cases where there was effective carbon sequestration over 30 or 40 years after different stages of accumulation and equilibrium [80,81]. Other research, based on modeling the impacts of land use change on SOC, had demonstrated potential carbon sequestration for even longer periods, for example, 50 or 100 years [82]. Furthermore, these authors highlighted the important role that the conversion of conventional agriculture to non-tillage could achieve. It is recognized that reducing emissions coming from agriculture is imperative [81] and sustainable agricultural practices have a significant influence on SOC sequestration [59], hence, changes to NT systems could make up for the total carbon emissions of agriculture globally, especially if conserving SOC is a "least-cost opportunity" for climate change adaptation and mitigation.

Natural ecosystems and managed grasslands and forests show a marked horizontal distribution of SOC and other variables. This is also frequently observed in previously degraded croplands which have been restored using conservation agriculture, in particular, NT management practices [75]. For this reason, the degree of stratification is considered an index of soil improvement and soil quality [55]; this is due to the positive effect of surface organic matter for erosion control, water infiltration, and the conservation of nutrients. With this in mind, the NT treatment produced the highest carbon stratification rate (SRSOC), which was 2.32 times higher than under CT, and in turn, SR for MT treatment was 1.36 times higher than in CT (Table 2). Consequently, in this specific context, NT can be regarded as a management practice that improves soil quality and provides a lower susceptibility to erosion.

According to Peregrina et al. [83], enrichment rates greater than two for SOC indicate improvements in soil quality. This limit was only exceeded under the NT treatment. Barbera et al. [47] reported that the annual sequestration of SOC with wheat monocropping was 2.75 times higher than with a wheat-faba bean rotation. However, in this study, considering the data available from the last years (2016–2017), and after 32 years, we did not find significant differences in SOC between wheat monocropping and wheat-vetch rotation.

### 4.2. Changes in Physical and Chemical Properties

The accumulation of vegetation residues on the topsoil is greater for the NT treatment; such organic material is a source of SOC, N, P, and K for the soil. This accumulation leads to the stratification found between topsoil (0–10 cm) and the layer underneath (10–20), which was remarkable for different tillage treatments. As expected, CT impeded stratification, and SR was not observed in this treatment, as SR values were 1.01, 1.16, 0.92, and 1.21 for SOC, N, P, and K respectively. On the contrary, NT treatment favored stratification, and SR values were 2.32, 1.96, 1.80 and 1.78 for these variables. The effects of MT were in between.

The pattern of concentrations found for SOC and total N were similar due to the high correlation between these variables, as usually organic forms of N represent more than 90% of total N in soils [84]. In addition, plant activity includes translocation of nutrients from deeper soil layers to the topsoil, these nutrients are not totally used by the crop near the soil surface [85], in this case from 0 to 10 cm.

Changes were also evident for P; the MT and NT treatments showed higher concentration of P and higher stratification compared to CT. It is well known that the availability of this nutrient is limited in calcareous soil suffering high fixation and little penetration [86] and hence high stratification.

Concerning K, it was significantly higher in the topsoil of NT treatment; this available form is bound to the surface of clays and humic substances and may be exchanged with other cations. In this semi-arid environment, the movement of K is limited by low moisture [87], and consequently, lixiviation is not favored, which leads again to stratification in less intensive tillage practices.

Soil structure plays a critical role in soil functions and can be measured using aggregate stability tests [88,89], which in turn can be used as soil quality indicators [90,91]. The CND test is sensitive to soil management practices [92]. In this study, the CND test values gradually decreased with increasing intensity of tillage systems following the order of macroaggregate resistance, NT > MT > CT, which coincided with gradual decreases in SOC content.

The CND test values changed with different soil tillage practices [27], but tests considering other structural properties, such as WSA (which defines the microaggregate stability), were not sensitive to these tillage practices. The aggregate stability also influenced, either directly or indirectly, other physical and chemical properties of the soils, so it can also be used as an indicator of soil degradation [90]. The macroaggregate stability was related to SOC and its protection from the activity of microorganisms [88]. Since roots and hyphae stabilize macroaggregates, macroaggregation is also affected by soil management. On the contrary, the water stability of microaggregates is strongly conditioned by persistent organic binding agents and seems to be independent of management [93]. Microaggregate stability was also related to clay content and probably to cationic links [94] that may be important in these argillo-calcareous soils.

Both TP and BD were highly influenced by tillage treatment and SOC content [23,95]. Usually, CT reduces BD in the topsoil [4,14]. In this study, BD increased under NT treatment. Similar observations have been made by other studies, for example, Costa et al. (2015) [96] . Increased BD was related to a decreased percentage of macro and mesoporosity. The lack of rotation exacerbates this influence, and BD was lower under the CT wheat monoculture treatment (1.36 Mg m$^{-3}$) and higher under the NT monoculture (1.51 Mg m$^{-3}$) treatment. Soil compaction has been frequently mentioned as one negative effect in soils under semiarid climates [97]. These differences were not as strong in plots under rotation, where there were no significant differences in BD between tillage treatments. Other variables followed the same pattern, which leads us to conclude that the effect of rotation (vetch-wheat in this study) may cushion, and even cancel out, the effects of different tillage systems in these particular soils.

*4.3. Changes in Water Infiltration and Water Availability*

Increased macroporosity as a result of CT is expected to influence the movement of water through soil [98,99]. However, several authors have found that NT systems yield higher water infiltration than CT systems [100,101]; other meta-analysis found that no-till did not consistently improve infiltration rates [102]. In this study, under the NT treatment, there was less infiltration than under the CT or MT treatments, despite the few differences in porosity. Infiltration tests were conducted in April 2017, a particularly dry year, during the period from January to April, when only 77 mm of rain was recorded in the area (60% lower than average according to the National Meteorological Agency records). Owing to the drought, the soil had multiple and large cracks which may overestimate infiltration rates. Under the NT system, soils had fewer cracks and therefore less infiltration than observed under the CT and MT systems. Vetch root systems may be less efficient than wheat root systems in creating effective connectivity between pores. Therefore, wheat-vetch rotations can hamper infiltration, when compared to monoculture in the local conditions studied here. The dry meteorological conditions during the experimental tests could have influenced these results; further research is needed to ascertain this hypothesis.

Under monocropping, AWC was higher under the NT treatment and lower under the CT treatment ($p < 0.05$; Table 3) owing mainly to the increased water content at field capacity. This suggests that in rainy years, soils managed under NT can hold more water than soils under CT.

Soils under NT showed PWP values higher than under other tillage systems. This can be a disadvantage in dry years, therefore, it may be a problem for adaptation under future scenarios of climate change. This was not observed in plots with rotations, therefore, we

can infer then that crops managed by rotation may be able to adapt more efficiently under future climate change.

When considering the ratio of AWC and TP (AWC/TP), we found that values were higher (although not significantly different) under the MT and NT treatments, compared to those under the CT treatment. This may indicate that under the CT treatment soils have a lower percentage of porosity able to hold water, which again, may be related to SOC content under moderate or low-intensity tillage systems. Water availability can be improved by NT systems in spite of the increase of bulk density, which may be due to the development of a new pore-size distribution under NT compared to CT [103]. We observed that CT reduced AWC if soils were managed using monocropping, but this negative effect was not observed in treatments with rotations. In this later case, rotation and NT dry soils increased the water retained at the permanent wilting point and this reduced AWC.

The NT may not be always beneficial under all soil types and climate scenarios [104,105] and should be studied under local conditions, particularly in semi-arid conditions [106].

## 5. Conclusions

The agricultural soils managed using no-tillage (NT) in central Spain increased their soil organic carbon (SOC) content for 32 years. These changes in SOC are significant for the upper 10 cm of topsoil but are less evident from the 10 to 20 cm depth; such a difference defined as SOC stratification is a positive indicator related to soil conservation. Due to this change, the NT treatment improved soil properties related to structure and porosity and showed higher available water capacity (AWC).

However, it has been argued that NT may reduce yield in dry years (in which soil moisture does not reach the field capacity) because these NT soils held more water at wilting point compared to those managed under minimum tillage (MT) or conventional tillage (CT). To improve water availability, it is essential that NT systems are combined with crop rotations, as under this combined management, no significant differences were found in AWC. Another negative effect of NT was the decrease in infiltration, which can be important in this semi-arid climatic condition.

The effects of tillage intensity were more pronounced under monocropping treatments, e.g., bulk density, SOC, field capacity, permanent wilting point, and AWC. This suggests that wheat-vetch rotations may mitigate the negative effects of tillage on soil structure, stability, and water holding capacity.

Under the MT system, most of the soil parameters had intermediate values compared to those recorded under the other two tillage systems.

Further long-term studies are needed in other different areas to confirm the effects of management practices on soil quality and water availability, to persuade growers to adopt more sustainable techniques as an alternative to CT in semi-arid areas and to determine the advantages of combining NT with rotations.

**Author Contributions:** Conceptualization, R.B., J.L.H., and V.S.-G.; methodology, R.B., B.S., A.G.-D., I.E., O.A. and M.J.M.; software, R.B., B.S., A.G.-D., and M.J.M.; resources, R.B. and B.S.; data curation, R.B., A.G.-D., L.N., M.J.S.d.A., O.A. and R.A.; writing—original draft preparation, R.B., B.S., A.G.-D., and M.J.M.; writing—review and editing, R.B., B.S., A.G.-D., R.A., and M.J.M.; supervision, R.B.; project administration, R.B., L.N., and M.J.S.d.A.; funding acquisition, R.B. and B.S. All authors have read and agreed to the published version of the manuscript.

**Funding:** The authors wish to thank the "Comisión Interministerial de Ciencia y Tecnología" (CICYT) of the Spanish Ministry of Education and Science for its financial support over 30 years of study (Grant Nos: PR84-0495; AGR90-0088; AGF96-1138-C02-01; AGL2001-3822-C02-01; AGL2002-04186-C03-01; AGL2007-65698-C03-01 and AGL2017-83325-C4-1R).

**Institutional Review Board Statement:** Not applicable.

**Informed Consent Statement:** Not applicable.

**Data Availability Statement:** Data supporting reported results can be found under request.

**Acknowledgments:** Thanks should be given to IMIDRA for their support during the last 30 years of field experimentation at "El Encín" experimental Station and the project AGRISOST-CM (S2013/ABI-2717) co-funded by the ESIF and Madrid Regional Government Education Office.

**Conflicts of Interest:** The authors declare no conflict of interest.

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
