# Peer review of "Tracking Changes on Soil Structure and Organic Carbon Sequestration after 30 Years of Different Tillage and Management Practices"

_agronomy, doi:10.3390/agronomy11020291_

Round 1

Reviewer 1 Report

Authors--your abstract is especially well done--clear, concise, readable.  Your methods were clear and convincing.  I did go back to them when I got to Table 3, for two reasons:  i) I was not familiar with your statistical method, and ii) I was uncertain about the field layout of your treatments.  If I read correctly, your rotational treatments were the largest block, with three reps of tillage treatments embedded inside each rotational treatment.  I am unfamiliar with that treatment approach, and was confused about how that could be the case.  Please clarify.  In Table 2, a modest issue ("vetch" appears to have been cut out of your header) and a more substantive one.  The larger issue (while not a game-changer) was that your letter system for means comparison seems to be within either tillage comparisons or rotational comparisons (rather than across those differing variables).  Please clarify this--perhaps simply by adding to the pertinent footnote.  Table 3 was my most important concern.  Since the vast majority of the results (germane to T3) focused on tillage systems within cropping systems, and since the multiple lettering system for treatment ranking is difficult to read (at least for this reviewer), perhaps the upper case ranking letters could be deleted, and the relatively few significant comparisons marked out currently by those upper case letters could simply be presented in your text.  Your discussion seems sound and well written.  I was unsure about the conclusion statement in lines 438-440 ("To avoid this disadvantage" sentence).  Not sure that your data support this assertion--maybe I missed something from your discussion regarding this point.

All in all, a strong paper and an excellent contribution to the literature.

Author Response

Responses to Reviewer 1

Dear Reviewer,

Thank you for your time and constructive suggestions. We are adding in blue the responses to your recommendations or requests. The Lines are referred to the document Word included “to track changes” at your convenience.

-----

Reviewer comment: Authors--your abstract is especially well done--clear, concise, readable.  Your methods were clear and convincing.  I did go back to them when I got to Table 3, for two reasons:  i) I was not familiar with your statistical method, and ii) I was uncertain about the field layout of your treatments.  If I read correctly, your rotational treatments were the largest block, with three reps of tillage treatments embedded inside each rotational treatment.  I am unfamiliar with that treatment approach, and was confused about how that could be the case.  Please clarify. 

This is what we meant, in order to clarify the design we change the wording and add (Line 109):

“Resulting in the experimental a design of three repetitions of 2 rotations × 3 tillage treatments.”

We do not add this table because other referees in a previous version thought that it was redundant.

Schematic chart of the experimental design

                                  Blocks

Soil Management

1

2

3

Conventional  Tillage:    CT

W - W

W - W

W - W

W - V

W - V

W - V

Minimum  Tillage:  MT

W - W

W - W

W - W

W - V

W - V

W - V

No-Tillage:  NT

W - W

W - W

W - W

W - V

W - V

W - V

W – W = Wheat-Wheat (monoculture); W – V (rotation) = Wheat-Vetch

In Table 2, a modest issue ("vetch" appears to have been cut out of your header)

The word vetch is now included, thank you.

… and a more substantive one.  The larger issue (while not a game-changer) was that your letter system for means comparison seems to be within either tillage comparisons or rotational comparisons (rather than across those differing variables).  Please clarify this--perhaps simply by adding to the pertinent footnote.  Table 3 was my most important concern.  Since the vast majority of the results (germane to T3) focused on tillage systems within cropping systems, and since the multiple lettering system for treatment ranking is difficult to read (at least for this reviewer), perhaps the upper case ranking letters could be deleted, and the relatively few significant comparisons marked out currently by those upper case letters could simply be presented in your text. 

In order to solve this issue, we have slightly changed the table caption. And, regarding your comments about multiple lettering, I understand the difficulties to appraise the differences considering lower and uppercase letters. We appreciate the comment to improve clarification and readability. Because the differences between rotation with the same tillage system were not frequent (just found for Macro and Mesoporosity, and Available Water Capacity), we can simplify the table deleting the uppercase letters, and adding only an asterisk to these few cases. E.g. the mesoporosity was higher for conventional tillage when rotation was applied, as it was 20.5% (CT+rotation VW) compared to 15.5% (CT+monocrop), in that case, we add *, please see Table 3. Only 6 pairs of asterisks are needed.

(Line 273) We slightly modify the table caption and add: “any differences found between rotation systems under the same tillage system are marked by an asterisk (*)

Your discussion seems sound and well written.  I was unsure about the conclusion statement in lines 438-440 ("To avoid this disadvantage" sentence).  Not sure that your data support this assertion--maybe I missed something from your discussion regarding this point.

(Line 488) We change the wording: “To improve water availability, it is essential that NT systems…”

All in all, a strong paper and an excellent contribution to the literature.

Thank you!

Reviewer 2 Report

The manuscript present interesting results concerning the soil quality and carbon sequestration using different tillage (no tillage) and management practices for long time of experimentation. Likewise, the authors highlights in their study the difference between the crop rotation and the monocropping. The subject of this work is interesting but there are some points that need to revise:

  • In the abstract, author cited in line 20, for first time, SOC abbreviation without symbol explication.
  • Authors have used in their results historic data obtained from Hernanz et al. we suggest to Authors to add this information in Material and Methods with short explication of Hernanz et al study.  
  • In the Figure 1. Can author present the standard deviation of sampling.
  • Line 200-203 authors said that there was a gradual and significant increase in SOC in both the MT and NT treatments, which was faster in NT. However, there were no major changes in the 10-20 cm depth, and there were no significant differences between tillage treatments. We suggest to authors to added p values signification of these results in the figure or in the text.
  • In the Table 2 authors must explain the symbol of N, P, K in the figure legend. Correct the column of crop rotation wheat-vetch and correct N, no NT (line 227) in the figure legend.
  • In the Table 3 authors have a different n number of sampling, can authors explained why?
  • In Discussion, we suggest to authors to discussed the differences in soil nutrients (N, P, K) between different tillage in their discussion. The results of table 2 showed interesting difference of these elements in 0-10 cm.
  • Finally, authors said in the line 341, This change could make up for the total carbon emissions of agriculture in Europe. We think that agricultural practices can help a lot in reduction of CO2 emission and the protection of the environment. We suggest the authors to give to this advantage a little focus in their study (discussion–conclusion) and add details in this regard.

Author Response

Responses to Reviewer 2

Dear Reviewer,

Thank you for your time and constructive suggestions. We are adding in blue the responses to your recommendations or requests.

We use the Line numbers in the Word document to track changes at your convenience.

-------

The manuscript present interesting results concerning the soil quality and carbon sequestration using different tillage (no tillage) and management practices for long time of experimentation. Likewise, the authors highlights in their study the difference between the crop rotation and the monocropping. The subject of this work is interesting but there are some points that need to revise:

  • In the abstract, author cited in line 20, for first time, SOC abbreviation without symbol explication.
  •  
  • Thank you, we add the meaning Soil Organic Carbon (SOC)
  •  
  • Authors have used in their results historic data obtained from Hernanz et al. we suggest to Authors to add this information in Material and Methods with short explication of Hernanz et al study.  

We mention the study of Hernanz et al 2009 in Materials and Methods, as suggested:

    • Line 129 (in the document to track changes) “This experiment started in 1985 with the aim to evaluate changes in SOC, originally CT was compared with MT and NT, considering three different rotations systems: wheat monocropping, vetch-wheat, and pea (Pisum sativum)-wheat, the latter was changed by vetch in 2002 (Hernanz et al. 2009 [55]). This research is following the same experimental design albeit includes only plots having monocropping and wheat-vetch rotations over time.”
  • This sentence is added at the end of the Experimental Design to avoid confusion between treatments, as the first decade there were three rotation systems.
  • All the references have been renumbered from 56 to the end accordingly.
  • In the Figure 1. Can author present the standard deviation of sampling.

The standard deviations have been added as requested.

  • Line 200-203 authors said that there was a gradual and significant increase in SOC in both the MT and NT treatments, which was faster in NT. However, there were no major changes in the 10-20 cm depth, and there were no significant differences between tillage treatments. We suggest to authors to added p values signification of these results in the figure or in the text.
  • Thanks to the previous suggestion, the addition of standard deviations in Fig 1, we can see now that significant differences are found in the last period of study for the MT and NT treatments, as the standard deviations do not overlap. In the previous sentence, we mentioned that p<0.05, I think that there is no need to repeat it to improve readability.
  • We add (Line 213)“ as can be observed for the last period of study in Figure 1”
  •  
  • In the Table 2 authors must explain the symbol of N, P, K in the figure legend. Correct the column of crop rotation wheat-vetch and correct N, no NT (line 227) in the figure legend.
  • Changed as requested in the new version
  • In the Table 3 authors have a different n number of sampling, can authors explained why?

In this Table 3, “n” is different due to different number of test performed for each measurement. The usual number of replicates was 12, however:

For operational reasons, only 3 infiltrations were performed per treatment. Each one lasts from 3 to 4 hours and needs dozens of liters of water in an area having very limited availability.

For statistical reasons (little variability), only 6 replicates were performed with parameters related to porosity: bulk density, porosity (macro, meso and total) and water held at field capacity and permanent wilting point.

In order to clarify this aspect, we add, at the Materials and Methods, 2.4 Statistical analysis:

(Line 194) “The number of repetitions was different for different variables due to the diverse number of tests performed for each measurement.  Twelve repetitions were performed for physical-chemical variables, laboratory analysis were duplicates; six repetitions were performed for variables related to soil porosity, due to their little variability and 3 infiltrations tests were performed per treatment due to operational reasons.”

  • In Discussion, we suggest to authors to discussed the differences in soil nutrients (N, P, K) between different tillage in their discussion. The results of table 2 showed interesting difference of these elements in 0-10 cm.

We add the standard deviations of these parameters in table 2, and we add several comments:

Results section:

(Line 239) “Similarly, an increase of N was observed in NT treatments at 0-10 cm depth, in this case, the SR indicated that N content at the topsoil nearly doubled the value found at 10 to 20 cm depth (SR=1.96). The effect of rotation also showed an increasing trend although variability impeded statistical significance.

The content of P and K followed the same pattern, higher concentration at the surface (0 -10 cm), and higher concentration as tillage intensity decreased. This pattern yielded progressively higher SR for P and K due to tillage treatments, but it was not detected for rotation treatments. “

Discussion section:

(Line 389) “The accumulation of vegetation residues on the topsoil is greater for the NT treatment, such organic material is a source of SOC, N, P and K for the soil. This accumulation leads to the stratification found between topsoil (0-10 cm) and the layer underneath (10-20) which was remarkable for different tillage treatments. As expected, CT impeded stratification, and SR was not observed in this treatment, SR values were 1.01; 1.16, 0.92 and 1.21 for SOC, N, P and K respectively. On the contrary, NT treatment favored stratification and SR values were 2.32; 1.96, 1.80 and 1.78 for these variables. The effects of MT was in between.

The pattern of concentrations found for SOC and total N are similar due to the high correlation between these variables, as usually organic forms of N represents more than 90% of total N in soils[84] [1].

In addition, plant activity includes translocation of nutrients from deeper soil layers to the topsoil, these nutrients are not totally used by the crop near the soil surface [85][2], in this case from 0 to 10 cm.

Changes were also evident for P, the MT and NT treatments showed higher concentration of P and higher stratification compared to CT. It is well known that the availability of this nutrient is limited in calcareous soil suffering high fixation and little penetration [3] hence, high stratification.

Concerning K, it was significantly higher in the topsoil of NT treatment, this ava [86]ilable form is bound to the surface of clays and humic substances, and may be exchanged with other cations. In this semi-arid environment, the movement of K is limited by low moisture [87] [4], and consequently lixiviation is not favored, which leads again to stratification in less intensive tillage practices. “

  • Finally, authors said in the line 341, This change could make up for the total carbon emissions of agriculture in Europe. We think that agricultural practices can help a lot in reduction of CO2emission and the protection of the environment. We suggest the authors to give to this advantage a little focus in their study (discussion–conclusion) and add details in this regard.[4]

We agree, this is an important outcome of this research. Following this suggestion, we elaborate on this topic, adding the following paragraph. We change “Europe” by “globally”, as there is no reason to exclude other regions of the world. 

(Line 364)“It is recognized that reducing emissions coming from agriculture is imperative [81] and sustainable agricultural practices have a significant influence on SOC sequestration [59],  hence, changes to NT systems could make up for the total carbon emissions of agriculture globally. Especially, if conserving SOC is a “least-cost opportunity” for climate change adaptation and mitigation.”

We have been thinking about including a statement about CO2 emissions in the Conclusions, as suggested, but, considering that we have not measured emissions, we honestly think that it could be considered speculative in this particular section of the article.

  1. Kelley, K.R.; Stevenson, F.J. Forms and nature of organic N in soil. Fertil. Res. 1995, 42, 1–11, doi:10.1007/BF00750495.
  2. Deubel, A.; Hofmann, B.; Orzessek, D. Long-term effects of tillage on stratification and plant availability of phosphate and potassium in a loess chernozem. Soil Tillage Res. 2011, 117, 85–92, doi:https://doi.org/10.1016/j.still.2011.09.001.
  3. Rhoades, H.F. Effect of Organic Matter Decomposition on the Solubility and Fixation of Phosphorus in Alkaline Soils; Nebraska, 1939;
  4. Raghad, M.; Alsaede, A.; Iqbal, M. Behavior of Potassium in Soil: A mini review. Chem. Int. 2016, 2, 58–69, doi:DOI: 10.13140/RG.2.1.4830.7041.

All the references have been renumbered from 88 to the end accordingly

Reviewer 3 Report

The study addressed changes in important soil quality traits over a very significant period of conservation tillage.The manuscript was structured in a simple but effective way to offer a comprehensive picture of the effects of conservation tillage systems over a considerable period of time and in different rotations. The effects of conservation tillage, combining intensity and methods, are a relevant finding of the study and this is appropriately focused on in the discussion and conclusion.

Author Response

Thank you for your time and for your positive review!

We have performed a few minor changes suggested by the other two reviewers to improve the first version, but no significant changes have been done.

Kind regards

The authors.